# ItNet: iterative neural networks for fast and efficient anytime prediction

## Abstract

Deep neural networks have usually to be compressed and accelerated for their usage in low-power, e.g. mobile, devices. Common requirements are high accuracy, high throughput, low latency, and a small memory footprint. A good trade-off between accuracy and latency has been shown by networks comprising multiple intermediate outputs. In this study, we introduce a multi-output network that has a tiny memory footprint in terms of its computational graph, which allows its execution on novel, massively-parallel hardware accelerators designed for extremely high throughput. To this end, the graph is designed to contain loops by iteratively executing a single network building block. These so-called iterative neural networks enable state-of-the-art results for semantic segmentation on the `CamVid` and `Cityscapes` datasets that are especially demanding in terms of computational resources. In ablation studies, the improvement of network training by intermediate network outputs as well as the trade-off between weight sharing over iterations and the network size are investigated.

## 1 Introduction

For massively-parallel hardware accelerators (Schemmel et al., 2010; Merolla et al., 2014; Yao et al., 2020; Graphcore, 2020a), every neuron and synapse in the network model has its physical counterpart on the hardware system. Usually, by design, memory and computation is not separated anymore, but neuron activations are computed next to the memory, i.e. the parameters, and fully in parallel. This is in contrast to the rather sequential data processing of CPUs and GPUs, for which the computation of a network model is tiled and the same arithmetic unit is re-used multiple times for different neurons. Since the computation is performed fully in parallel and in memory, the throughput of massively-parallel accelerators is usually much higher than for CPUs and GPUs. This can be attributed to the fact that the latency and power consumption for accessing local memory, like for in-memory computing, are much lower than for computations on CPUs and GPUs that require the frequent access to non-local memory like DRAM (Sze et al., 2017). However, the network graph has to fit into the memory of the massively-parallel hardware accelerators to allow for maximal throughput. If the network graph exceeds the available memory, in principle, the hardware has to be re-configured at high frequency as it is the case for CPUs and GPUs and the throughput would be substantially reduced. Mixed-signal massively-parallel hardware systems usually operate on shorter time scales than digital ones, e.g. compare Schemmel et al. (2010) and Yao et al. (2020) to Merolla et al. (2014) and Graphcore (2020a), and would allow for even higher throughputs.

In order to achieve neural networks with tiny computational graphs, in which nodes are operations and edges are activations, we heavily re-use a single building block of the network (see the iterative block in Figure 1a). Not only the structure of computations, i.e. the type of network layers including their dimensions and connectivity, is identical for each iteration of this building block, but also the weights are shared between iterations. In the computational graphs of these so-called *iterative neural networks* (ItNet), the re-used building blocks with shared weights can be represented by nodes with self-loops. Compared to conventional feed-forward networks, loops simplify the graph by reducing the number of unique nodes and, consequently, its computational footprint. However, the restriction of sharing weights usually decreases the number of free parameters and, hence, the accuracy of networks. To isolate and quantify this effect we compare networks with weight sharing to networks, for which the parameters of the building blocks are chosen to be independent between iterations of

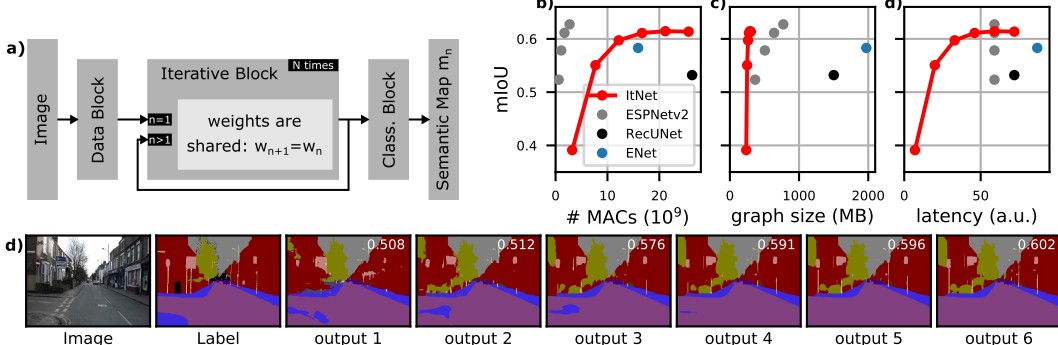

Figure 1: This study in a nutshell. **a)** The iterative neural network (ItNet): first, images are pre-processed and potentially down-scaled by a sub-network called *data block*. Then, the output of this data block is processed by another sub-network that is iteratively executed. After every iteration $n$, the output of this *iterative block* is fed back to its input and, at the same time, is further processed by the *classification block* to predict the semantic map $m_n$. This network generates multiple outputs $m_n$ with increasing quality and computational costs and heavily re-uses intermediate network activations. While the parameters are shared between iterations of the iterative block, the parameters of the classification block are independent for each $n$. **b-d)** Intersection-over-union (mIoU) over multiply-accumulate operations (MACs), the size of the computational graph, and the latency, respectively, on the validation set of the `Cityscapes` dataset. To our knowledge ESPNetv2 (Mehta et al., 2019) is the state-of-the-art in terms of mIoU over MACs, for which we show the mIoU if not using pre-trained weights (Ruiz & Verbeek, 2019). Wang et al. (2019) share the weights of U-Nets and recurrently connect their bottlenecks. Note that the ItNet requires more MACs, but has a substantially smaller size of the computational graph, and a lower latency than the reference networks. The throughput of ItNets is potentially also higher, since the small graph size allows for the execution on massively-parallel hardware systems. **e)** Image, label, and network predictions (with the mIoU in their corner) over the network outputs of the ItNet in (c). The data sample with the 75th-percentile mIoU is shown.

the building block. In contrast to the above proposal, conventional deep neural networks for image processing usually do not share weights and have no (e.g. Huang et al., 2017) or few (e.g. one building block for each scale like by Greff et al., 2017) layers of identical structure. Liao & Poggio (2016) share weights between re-used building blocks, but use multiple unique building blocks.

To improve the training of networks, which contain loops in their graphs, and to reduce the latency of networks during inference we use multiple intermediate outputs. Multi-output networks that heavily re-use intermediate activations are beneficial for a wide range of applications, especially in the mobile domain. In an online manner, they allow to trade off latency versus accuracy with barely any overhead (e.g. Huang et al., 2018). From an application point of view, the benefit of this trade-off can be best described in the following two scenarios (Huang et al., 2018): In the so-called *anytime prediction* scenario, the prediction of a network is progressively updated, whereas the first output defines the initial latency of the network. In a second scenario, a limited computational budget can be unevenly distributed over a set of samples with different "difficulties" in order to increase the average accuracy.

Since all nodes in the network graph are computed in parallel on massively-parallel hardware systems (e.g. Esser et al., 2016), the latency for inference is dominated by the depth of the network, i.e. the longest path from input to output (Fischer et al., 2018). Consequently, we prefer networks that compute all scales in parallel (similar to Huang et al., 2018; Ke et al., 2017) and increase their depth for each additional intermediate output to networks that keep the depth constant and progressively increase their width (Yu et al., 2019). Furthermore, multi-scale networks are also beneficial for the integration of global information, as especially required by dense prediction tasks like semantic segmentation (Zhao et al., 2017). To further reduce the latency we also reduce the depth of the building blocks for each scale.

In deep learning literature, the computational costs are usually quantified by counting the parameters and/or the multiply-accumulate operations (MACs) required for the inference of a single sample. For fully convolutional networks, the number of parameters is independent of the spatial resolution

of the network's input and the intermediate feature maps. Especially for large inputs as commonly used for semantic segmentation, the number of parameters does not cover the main workload and is, hence, not suited as a measure for computational costs. MACs have the advantage that they can be easily calculated and are usually a good approximation for the latency and throughput on CPUs and even GPUs. However, for most novel hardware accelerators, not the MACs, but memory transfers are dominating the computational costs in terms of power consumption (Chen et al., 2016; Sze et al., 2017; Chao et al., 2019). These memory transfers are minimized on massivly-parallel hardware systems as long as the network graph fits into the *in-computation memory* of these systems, i.e. the memory of their arithmetic units. Since both the power consumption during inference and the production cost scale with the size of this memory, in addition to MACS, we also compare the size of the computational graphs between networks. Note that the practical benefits of ItNets cannot be demonstrated on conventional CPUs or GPUs, since these hardware systems do not support the processing of neural networks in a fully-parallel and, hence, low-latency fashion.

Note that, in this study, we focus on network models in the low-power regime of only few billion MACs, while processing large-scale images as commonly used for semantic segmentation (for datasets, see Section 2.4). The key contributions of this study are:

- We introduce efficient networks with tiny computational graphs that heavily re-use a single building block and provide multiple intermediate outputs (Sections 2.1, 2.2 and 2.5).
- We search for the best hyperparameters of this model and investigate the effect of multiple outputs and weight sharing on the network training (Sections 2.3, 3.1 and 3.2).
- To our knowledge, we set the new state-of-the-art in terms of accuracy over the size of the computational graph and discuss the potential benefits for low-power applications.

We will release the source code upon acceptance for publication.

## 2 METHODS

The following networks process images of size $x \times y \times 3$ and output $N$ semantic maps $m_n$ of size $x \times y \times C$ with $C$ being the number of classes.

### 2.1 NETWORK ARCHITECTURE

We are interested in network architectures with a small computational footprint facilitating their application in mobile devices. To this end, we design a neural network that heavily re-uses the intermediate activations and weights (Figure 1a). Conceptionally, the network model can be split into three main building blocks: the *data block*, the *iterative block* and the *classification block* (for an overview and details, see Figure 1a and Figure 2, respectively). While the data block is executed only once for each image, the iterative block can be executed multiple times in a row by feeding back its output as the input for the next iteration. The classification block outputs the prediction of the semantic map by processing the intermediate activations of the feedback signal. While the weights of the iterative block are shared between iterations, the weights of the classification block are unique for each iteration. We consider the size of the computational graph and the number of MACs as meaningful indicators for the computational footprint (see also Section 1). To obtain a high-accuracy network architecture under these objectives we reduce the size of the computational graph by introducing loops and optimize the following architectural hyperparameters: the number of scales $L$, the number of iterations $N$, and the number of bottleneck residual blocks $K$.

### 2.2 NETWORK TRAINING

For training, we use a joint cost function for all outputs of the network:

$$\mathcal{L} = \sum_n \bar{a}_n c_n(\tilde{m}_n, m_n),$$

where $c_n$ is the categorical cross entropy between the true labels $\tilde{m}_n$ and the network predictions $m_n$. The weight factors $a_n$ are normalized as follows: $\bar{a}_n = a_n/(\sum_i a_i)$.

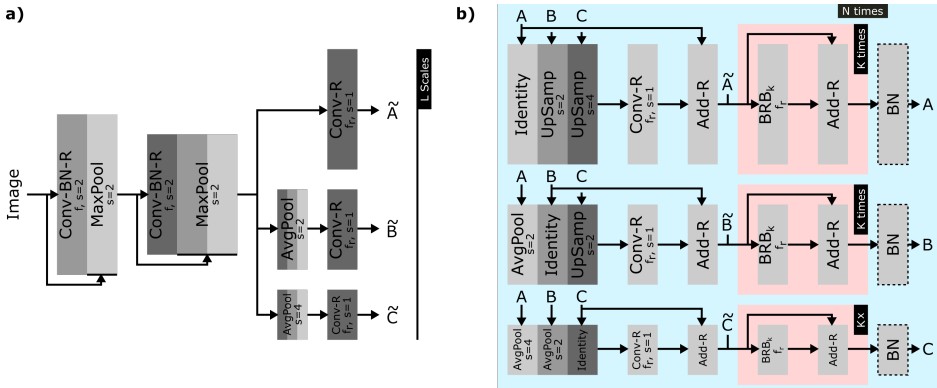

Figure 2: Detailed description of the *data block* (a) and *iterative block* (b) as shown in Figure 1:
**a)** First, images are down-sampled in two stages, each using a stride and pooling of size $s = 2$
(inspired by Paszke et al., 2016). Then, the output of the second stage is processed on $L$ different
scales. For clarity, an example with $L = 3$ is shown. Convolutional layers (with kernel size 3), batch-
normalization layers and ReLU activation functions are denoted with `Conv`, `BN` and `R`, respectively.
**b)** The iterative block receives input at $L$ different scales denoted as `A`, `B` and `C`. In order to mix the
information between the different scales (e.g. Zhao et al., 2017) the inputs `A`, `B` and `C` are concatenated
and processed with a residual block (He et al., 2016). Then, an additional $K$ *bottleneck residual
blocks* (BRB; with expansion factor $t = 6$; Sandler et al., 2018) are applied to obtain the output of
the iterative block consisting of feature maps at again $L$ different scales. This output is both fed back
as input for the next iteration $n$ and forwarded to the classification block. The classification block
consists of a $1 \times 1$ convolutional layer with $C$ output channels and an 8-fold bilinear up-sampling
to the original spatial dimensions. In the first iteration ($n = 1$) of the iterative block, the mixing of
the scales is skipped and the input is directly processed by the $K$ bottleneck residual blocks (see
injection of input as Ã, B̃ and C̃). The number of output channels are denoted by $f$ and $f_r = f/2$,
respectively. The batch-normalization layers within the iterative blocks (dashed boxes) are never
shared between iterations.

We use the Adam optimizer with $\beta_1 = 0.9$, $\beta_2 = 0.999$ and a learning rate $0.001$ that we multiply
with $0.1$ after $70\%$ and $85\%$ of the number of overall training epochs. We use a batch size of $8$ and
train the network for $2000$ ($4000$ for Figure 5) and $900$ for the `CamVid` and `Cityscapes` datasets,
respectively. For Figure 3, Figure 4 and the appendix, we report the mean values and the errors of the
means across $5$ trials. For Figures 1 and 5, we report the trial with the highest peak accuracy over $3$
trials.

For the results shown in Figures 1 and 5, we use dropout with rate $0.1$ after the depth-wise convolutions
in the bottleneck residual blocks and an L2 weight decay of $10^{-5}$ in all convolutional layers. For all
other results, we do not use dropout and weight decay.

## 2.3 Network evaluation

Throughout this study, we measure the quality of semantic segmentation by calculating the mean
intersection-over-union (mIoU Jaccard, 1912), which is the ratio of the area of overlap and the area
of union

$$IoU = \frac{label \cap prediction}{label \cup prediction}$$

averaged over all classes.

We consider a network to perform well if it achieves a high mIoU while requiring few MACs. To this
end, we calculate the area under the curve of the mIoU ($y_n$) over MACs ($x_n$) with output index $n$ as
follows:

$$\overline{AUC} = \sum_{n=0}^{N-1} (x_{n+1} - x_n) \left( \frac{(y_{n+1} - y_0) + (y_n - y_0)}{2} \right)$$

with $(x_0, y_0) = (0, 0.00828)$, where $y_0$ denotes the mIoU at chance level for the `CamVid` dataset. To compensate for different maximum numbers $x_N$ of MACs for different sets of hyperparameters, we normalize as follows: $AUC = \frac{\overline{AUC}}{x_N}$.

The size of the computational graph is computed by accumulating the memory requirements of all nodes, i.e. network layers, in the network graph. For each layer, the total required memory is the sum of the memory for parameters, input feature maps and output feature maps.

The theoretical latency of a network if executed fully in parallel is determined by the length, i.e. the depth, of the path from input to output of this network (see also Section 1).

For both the size of the computational graph and the latency, we only consider convolutional layers like commonly done in literature (e.g. Paszke et al., 2016; Wu et al., 2018; Mehta et al., 2019). This means, we ignore other network layers like normalizations, activations, concatenations, additions and spatial resizing, for which we assume that they can be fused with the convolutional layers.

### 2.4 DATASETS

The `CamVid` dataset (Brostow et al., 2008) consists of 701 (367 for training, 101 for validation, 233 for testing) annotated images filmed from a moving car. We use the same static pre-processing as in Badrinarayanan et al. (2017) to obtain images and semantic labels of size $480 \times 360$ and normalize the pixel values to the interval $[0, 1]$ by dividing all pixel values by 255. For online data augmentation during training, we horizontally flip the pairs of images and labels at random and use random crops of size $480 \times 352$.

The `Cityscapes` dataset (Cordts et al., 2016) consists of 3475 (2975 for training, 500 for validation) annotated images and we use the validation set for testing. We resize the original images and semantic labels to $1024 \times 512$ and divide all pixel values by 127.5 and subtract 1 to obtain values in the interval $[-1, 1]$. For online data augmentation during training, we horizontally flip the pairs of images and labels at random.

For both datasets, pixels with class labels not marked for training are ignored in the cost function.

### 2.5 BATCH NORMALIZATION

In case of independent parameters between iterations of the iterative block, i.e. $w_{n+1}$ is independent from $w_n$ in Figure 1a, batch normalization improves the network training. However, in case of weight sharing, also sharing the parameters of batch normalization between iterations significantly worsens network training. Since not sharing batch-normalization parameters would violate our idea to re-use the identical building block again and again, we place batch-normalization layers between the iterations of the iterative block (see Figure 2b). For comparability, we use the same setup also for networks without weight sharing, although the average validation mIoU is slightly decreased compared to networks that instead use batch normalization after each convolution (Figure 3).

## 3 RESULTS

In order to obtain accurate networks with low computational costs, we first search for the best set of the architectural hyperparameters $L$, $N$ and $K$ (Section 3.1). Since we are also interested in the trade-off between weight sharing and the networks size, we also consider networks without weight sharing in this search. Then, we investigate the impact of intermediate losses on the network performance to find the best set of weight factors for the loss function (Section 3.2). Finally, for the found set of hyperparameters and weight factors, we show results of ItNets on the `CamVid` and `Cityscapes` dataset (Section 3.3).

### 3.1 SEARCH FOR ARCHITECTURAL HYPERPARAMETERS

In order to compare ItNets with and without weight sharing between iterations of the iterative block our goal is to find a common set of architectural hyperparameters for both scenarios. We perform a grid search over the hyperparameters of our network model as described in Figures 1 and 2. For each set of hyperparameters, we choose the largest possible number of channels $f$ that results in a network

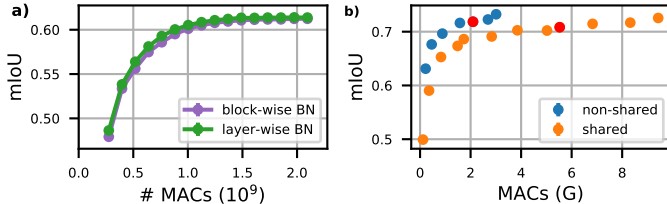

Figure 3: **a)** Comparison of mIoU over MACs between networks with batch normalization applied between iterative blocks (purple curve) and applied after each convolution (green curve) on the test set of the `CamVid` dataset. Note that, for both cases, batch normalization is always applied after each convolution in the data block. **b)** Peak mIoU over MACs for networks with different widths on the validation set of the `CamVid` dataset. Each data point represents the peak mIoU over all network outputs for one specific network width and the corresponding MACs for this output. For the following studies, we selected the network highlighted in red that achieves, out of 5 trials, a maximum mIoU of 72.7 and 72.8 for independent and shared weights, respectively.

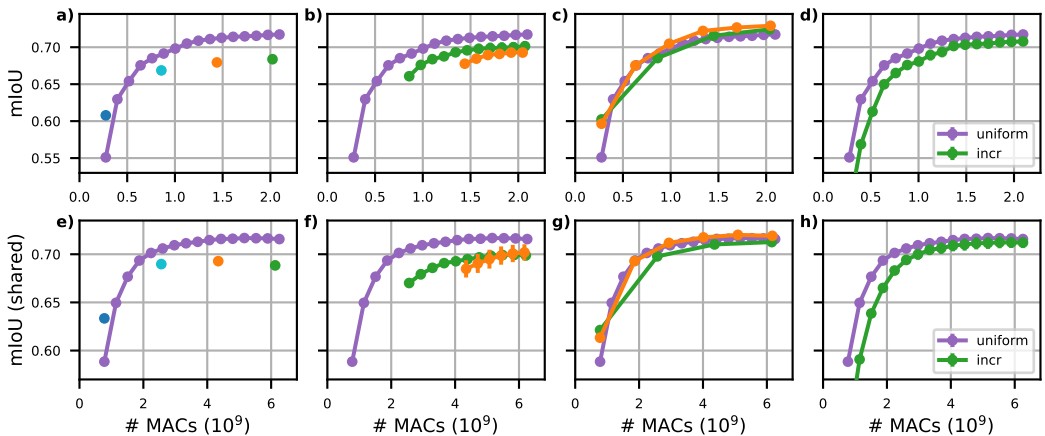

Figure 4: Ablation studies for multi-output training on the validation set of the `CamVid` dataset: The mIoU over MACs is shown for the same architectural hyperparameters, but for different sets of $a_n$. **a-c and e-g)** The weight factors $a_n$ are set to 1 for the shown outputs and to 0 otherwise. Each different set of weight factors is highlighted with a different color. **d and h)** The weight factors $a_n$ are modulated over the network outputs. We increase the weight factor from 1 to 16 (incr) or keep the weights uniform over all outputs.

with less than 2.2 billion MACs for the last output of the network on single samples of the `CamVid` dataset.

Since we are interested in network architectures with a high mean intersection-over-union (mIoU) and a low number of MACs, we sort the architectures by their area under the curve (see $AUC$ in Section 2.3) for both networks with independent and shared weights. Networks with a peak performance $\max_n(mIoU_n) \leq mIoU_{\text{totmax}} - 0.02$, where $mIoU_{\text{totmax}}$ is the highest peak performance over all sets of hyperparameters, are considered as not suitable for applications and are discarded. For comparability between networks with independent and shared weights, we choose the architecture with the lowest average rank across the two classes of networks and obtain the following set of hyperparameters: $N = 16$ iterations, $L = 3$ scales and $K = 1$ bottleneck residual blocks (for the results of the grid search, see Figures 6 and 7 in the appendix). Note that many iterations ($N = 16$) result in a high re-usage of the iterative block, a low latency and a small computational graph.

### 3.2 MULTI-OUTPUT TRAINING

We study the impact of additional intermediate network outputs on the network performance by training the same network architecture with different sets of weight factors $a_n$ of the loss function

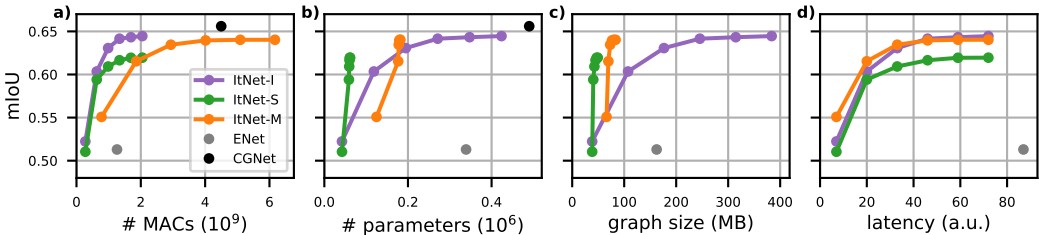

Figure 5: Network performance in terms of mIoU over MACs **(a)**, mIoU over number of parameters **(b)**, mIoU over memory bandwidth **(c)**, and mIoU over memory footprint of the intermediate state **(d)** on the test set of the `CamVid` dataset. In case of weight sharing, we increased the width of the network (compare orange to green curve) to recover the mIoU of the network with independent weights (blue curve). According to Figure 4c and g we thin out the losses. ENet (Paszke et al., 2016) and CGNet (Wu et al., 2018) denote efficient reference networks. Note that the MACs of these reference data points are normalized to the actual input size ($480 \times 360$) that was used to obtain the reported mIoU values.

(for details, see Section 2.2). In summary, later network outputs benefit from the additional loss applied to earlier outputs. This is supported by ablation studies, in which we trained networks with single, only late or thinned out outputs. The following results and conclusions are similar for both network types with and without weight sharing and, hence, we discuss them jointly.

The training of single outputs is worse than jointly training all outputs, except for only training the first output (see Figure 4a and e). However, a network with only the first output has no practical relevance due to its tiny network size and, consequently, low mIoU.

The intermediate activations of the early layers are optimized for the potentially conflicting tasks of providing the basis for high accuracy at early outputs and, at the same time, top accuracy at late outputs. However, our experiments suggest that especially the optimization of the early outputs improves the performance of late outputs (see Figure 4b and f). Our observation that early losses do not conflict with late losses allows to keep these early losses resulting in networks with very low latency (see Figure 5d). In addition, keeping the first output to allow for low latency, but decreasing the density of outputs, removes unnecessary constraints and slightly improves the mIoU (see Figure 4c and g). The removed outputs are likely not required in applications, since the improvement of the mIoU between consecutive outputs is rather small (see Figure 1e).

So far, we removed sets of outputs from the loss function, but kept the weight factors $a_n$ identical for all remaining outputs. For linearly increasing the weight factors over the outputs, the mIoU jointly decreases for all outputs (compare different colors in Figure 4d and h). This unexpected decrease in the mIoU for late outputs may be attributed to the observed effect that earlier outputs are crucial for late performance as discussed above.

## 3.3 THE COST OF WEIGHT SHARING

In the following, we address the cost of sharing weights in the iterative block. Specifically, we compare the number of MACs, the number of parameters, the size of the computational graph and the latency between networks with and without weight sharing.

Compared to networks with independent weights, networks with shared weights have substantially less free parameters that, consequently, results in worse mIoUs. To compensate for this decrease in the peak mIoUs we increase the width $f$ of the latter by a factor of $1.76$ ($f = 51$ with weight sharing compared to $f = 29$ without weight sharing), which corresponds to approximately three times the number of MACs (Figure 3b).

Although the number of MACs has to be increased for weight sharing (Figure 5a), the number of parameters is substantially smaller ($44\%$), since parameters are highly re-used (Figure 5b). The size of the computational graph is dominated by the size of the feature maps due to their large spatial dimensions. If the weights are shared between iterations of the iterative block the previously unrolled network graph can be collapsed into a single node with a self-loop reducing the size of the

computational graph (Figure 5c). The latency is identical for all networks, since we use the same architectural hyperparameters (Figure 5d).

## 4 Discussion

This study investigates the trade-off between the size of the computational graphs and the accuracies of so-called iterative neural networks that show state-of-the-art performance for this trade-off. Iterative networks require less than half the size for the computational graph compared to networks that are state-of-the-art in terms of the number of MACs (compare 276MB for ItNet to 639MB for ESPNetv2 to achieve an mIoU of 61.1 in Figure 1c). In addition, the multiple intermediate network outputs allow for any-time prediction and a lower latency that is further reduced by the shallow network design (Figure 1d). While targeting the same accuracy with an identical network structure, the sharing of weights reduces the size of the graph by a factor larger than 4 (compare 83MB to 384MB to achieve an mIoU of approximately 64.2 in Figure 5c), but increases the number of MACs by a factor of approximately 3 (compare 2.0 to 6.2 billion MACs in Figure 5a). However, this increase in MACs is expected to be smaller than the increase in throughput by novel, massively-parallel hardware accelerators, especially if the computational graphs fit into the in-computation memory of these hardware systems (see also Section 1). For example, Graphcore (2020b) reports a 300-fold increase in throughput compared to GPUs for recurrent networks of small size and Esser et al. (2016) report more than 5000 frames per second and Watt for computer vision tasks. During the training of ItNets, we observe that additional and especially early network outputs improve the performance of late outputs, which was also observed by Zhou et al. (2019) on pre-trained networks for age estimation. In principle, the presented methods could also be applied to different tasks, like object classification and detection, for which we expect similar observations. However, these datasets are usually provided in a lower spatial resolution, which reduces the challenge and the need to reduce the size of these already rather small computational graphs.

Since massively-parallel hardware systems are not easily accessible at the moment, we cannot provide real benchmark data, but will in the following exemplarily discuss the potential benefits of our network models if executed on such hardware systems. To this end, we describe the execution of our model on the Graphcore (2020a) system, since this system seems to be the most promising in terms of commercialization and availability. If the full network graph fits into the in-computation memory of the Graphcore (2020a) system (3.6GB for the `IPU-M2000` machine), the input data can be pipelined through this graph in a streaming fashion without the need to reconfigure the hardware. Then, by design, independent nodes in this graph are executed fully in parallel (see also Fischer et al., 2018). This results in substantially lower latencies during network inference compared to systems that do not execute network graphs fully in parallel like CPUs and GPUs (an up to 25 times lower latency is reported by Graphcore, 2020b). In contrast, for CPUs and GPUs, the workload of the network graph has to be tiled and the arithmetic units are continuously reconfigured for each tile. ItNets significantly reduce the footprint of their graphs by introducing loops and, hence, improve the network's accuracy for the same footprint. Although the in-computation memory is huge for the `IPU-M2000` machine that is optimized for data centers, we expect embedded versions of similar systems to have a considerably smaller in-computation memory that, then, require networks with tiny computational graphs. Other massively-parallel hardware designs increase the density of the memory in terms of synapses per chip area by aggressive quantization (Merolla et al., 2014) or mixed-signal implementations (Schemmel et al., 2010; Yao et al., 2020) and impose additional challenges for the development of network models.

For the `Cityscapes` dataset, our ItNet already requires two `Nvidia V100` GPU with 32GB memory if the standard training procedure is not modified. Consequently, exploring large-scale ItNets is difficult by using backpropagation on GPUs. However, since ItNets are optimized for their execution on massively-parallel hardware systems, this demonstrates the fundamentally different operation principles between these systems and GPUs as well as highlights the need to think beyond workloads that are tailored for GPUs.

In parallel to this study, a novel training technique was introduced by Bai et al. (2020) that significantly improves the scaling of vision networks composed of iteratively executed building blocks with weight sharing. Instead of using backpropagation, they optimize the equilibrium point of a fix-point process described by the iterative execution of the building block. They achieved task accuracies comparable

to state-of-the-art networks and, to this end, required approximately the same number of parameters compared to the baselines. In line with our study, this results in comparably large building blocks that are executed many times and, consequently, require significantly more MACs than the baseline networks. However, the training method of Bai et al. (2020) can not be applied to networks, for which the weights are allowed to be updated between iterations and, hence, the effect of weight sharing can not be studied in isolation. In addition, their method includes the raw image (8MB in `float32` for Cityscapes) into the state of the iterative block, which prevents to shrink their networks to the regime of few billion MACs as shown by our method, for which the state has a size of only 4.8MB.

The concept of recurrent building blocks has also been previously applied to semantic segmentation (Pinheiro & Collobert, 2014) and to generate super-resolution images (Cheng et al., 2018). However, these works share the same network building block between different scales of the feature maps, which does not allow for anytime predictions. Instead of processing all scales at once like in Bai et al. (2020) and the work at hand, a full encoder-decoder network can be shared over time. To predict semantic maps for single images (Wang et al., 2019) or videos (Valipour et al., 2017) the intermediate feature maps of this encoder-decoder network are connected by recurrent units over time. Ballas et al. (2016) apply a similar technique to networks with encoders only on video tasks.

As an extension of this study, partly releasing the constraint of weight sharing may allow for a significant reduction in the required size of the iterative block. To this end, a ratio between shared and free weights may be chosen close to, but smaller than one. This will require to reconfigure the network graph on hardware, which may be costly or not possible at all depending on the hardware system. An alternative approach to release constraints would be to replace the weights of the building block by a functions that modulates the weights over the iterations of the building block. Another interesting open question for further studies is the root cause for the observation that early outputs improve the performance of late outputs. This effect may be attributed to some kind of knowledge distillation and / or to the shortcut for the gradients from network output to input.

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

## A    APPENDIX

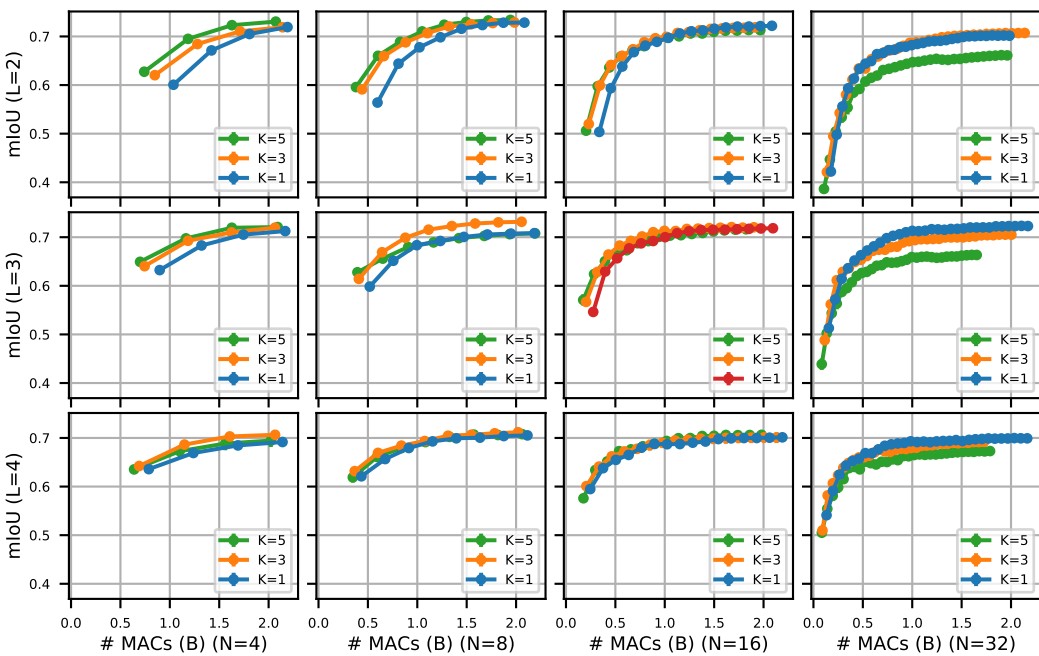

Figure 6:   Grid search over the following hyperparameters on the validation set of the `CamVid` dataset: number of layers $N$, number of blocks $L$ and number of residual blocks $K$. Parameters are not shared between iterations $n$ of the iterative block. The best set of hyperparameters (for details about the selection, see Section 3.1) is depicted in red.

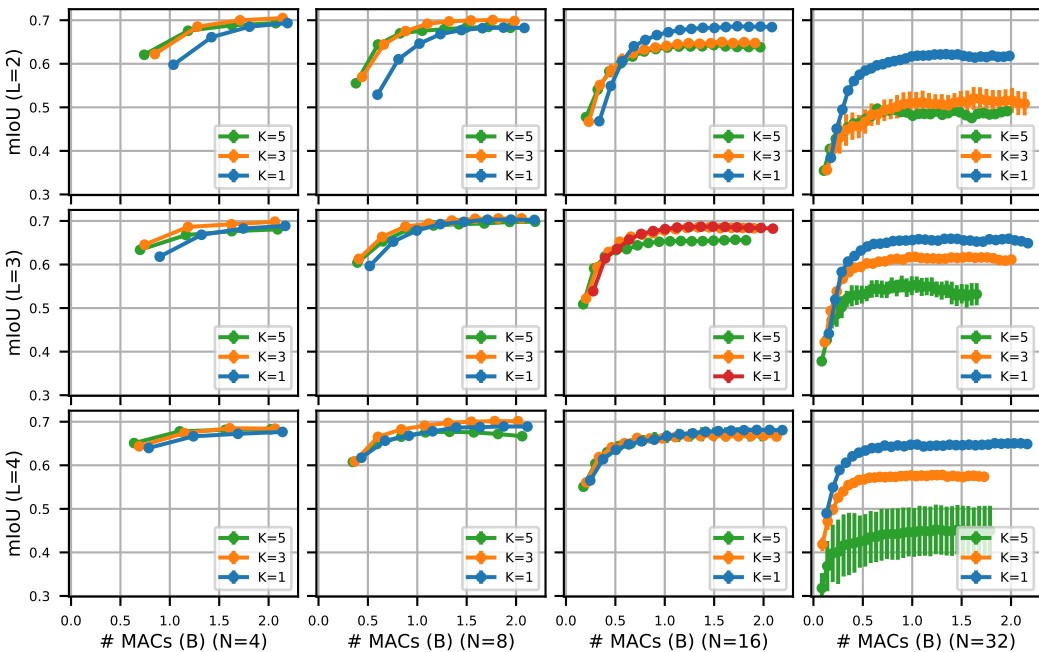

Figure 7:   Like Figure 6, but the parameters are shared between iterations $n$ of the iterative block.

