# OpenReview forum: "ItNet: iterative neural networks for fast and efficient anytime prediction"
_ICLR.cc/2021/Conference — Reject_

### Official Review · AnonReviewer1 · 2020-10-26
**A potentially interesting topic but studied on a too small scale**

**Rating:** 4
**Confidence:** 4

**Review:**

This paper studies the influence of the use of shared vs independent parameters in re-used blocks of neural networks. This is achieved for the task of semantic segmentation, with a network architecture that iteratively refines its prediction.

Strengths:
- Studying the effect of parameter sharing vs the use of independent parameters in recurrent types of architecture is interesting and could lead to a better understanding of these architectures
- The paper is clearly written, and the methodology would be reproducible


Weaknesses:

Contribution:
- While I do believe that such a study could provide the community with a better understanding of architectures with re-used blocks, the scale of the study performed in this paper is too small to draw conclusions. The paper tackles a single task (semantic segmentation), and, more critically, evaluates a single architecture. There is therefore no evidence that the conclusions drawn here will generalize to other architectures/tasks, which significantly limits the potential impact of this paper on the community.

Related work:
- While I am not aware of similar studies, several important references tackling the task of recurrent semantic segmentation are missing, e.g., Pinheiro & Collobert, ICML 2014; Wang et al., ICCV 2019. These methods rely on different architectures, and studying the effect of parameter sharing in their frameworks would broaden the scope of this work.
- Similarly, video segmentation has also been tackled with recurrent architectures, e.g., Ballas et al., ICLR 2016; Valipour et al., WACV 2017. Studying the use of parameter sharing in this context would thus also increase the potential impact of this work.

Empirical results:
- While the results indeed evaluate the effect of parameter sharing on the chosen architecture, they seem to be somewhat disappointing in terms of absolute performance. In particular, on CityScapes (Fig. 1(b)), CNMM yields significantly higher mIoUs than the proposed method for the same number of MACs. Wouldn't it be possible to also extend this study to the CNMM architecture?

Minor comments:
- In the caption of Fig. 1, the authors mention that the parameters of the classification block are never shared. Does this mean that there is one classification block for each recurrent iteration?
- In Section 2.2, when explaining the normalization of the weight factors, I suggest using a different notation for the weights before and after normalization, e.g., w_n = w_n'/(\sum_i w_i').
- The title of Section 3.1 (Neural Architecture Search) is a bit misleading, as the authors simply perform a grid search of a few hyper-parameters, not a proper search over a large space of different architectures as in the NAS sub-field.
- The meaning of the different colors in Fig. 4 is not always explicitly defined.

---

> ### Author Response · Authors · 2020-11-24
> **Addressing comments of reviewer 1**
>
> Thank you very much for these constructive, detailed and valuable comments that I will answer point for point below. For a summary of changes, see my comment above.
>
> >    Is the parameters of the re-usable building block updated iteratively during inference? The paper described two scenarios where the parameters of the re-usable building block can be updated or shared across multiple iterations. My first question is, is this for training time only or for both training and inference?
>
> This applies to both training and inference. We refined the introduction accordingly.
>
> >    a) At inference time if the parameter updates are allowed at each iteration, what is the mechanism to decide the new weights? If an independent block was used and only the structure of the building block is reused, there would not be any benefit for latency or memory footprint.
>
> To determine the cost of weight sharing the weights are updated at every iteration, since they are assumed to be completely independent during training. We agree that, in this case, the benefit for latency and memory footprint would be reduced. We revised the manuscript and now the case of weight sharing is the default case and we only discuss independent weights in the context of investigating the "costs" of weight sharing.
>
> >    b) If the parameters of the building blocks are fixed during inference, then it falls into the weight-sharing scenario where the model performance is much lower according to Figure 1 c).
>
> The isolation of this effect is one of the key contributions of this manuscript. We sharpened the list of contributions accordingly.
>
> >    Lack of anytime prediction experiments. The paper claims it studies anytime prediction setting, however, I couldn't find explicit descriptions about it in the experiment section and descriptions about how the model choose the number of iterations during inference.
>
> We follow the definition of "anytime prediction" as introduced by Huang (2018) that reads "anytime prediction, where the network can be forced to output a prediction at any given point in time". This means that the number of iterations has not to be chosen, but is defined by the available computational resources (see also Huang 2018).
>
> >    Missing Cityscapes experiments. The paper claims in the abstract that the method was evaluated on cityscapes but in the discussion section, it says they couldn't provide cityscapes experiments due to memory issues. The inconsistency in the paper should be fixed.
>
> We now provide a complete set of results for Cityscapes and updated the manuscript.
>
> >    What is the exact benefit of the homogenous network? The paper shows this structure requires at least 3 times larger MACs in order to achieve competitive performance. It also says this network has potential for novel, massively-parallel hardware accelerators. This is a bit vague to me, could you provide a concrete example?
>
> We added a concrete example for executing our model on a massively-parallel hardware system to the discussion. Although we currently have no access to such a system, we hope our claims are less vague.
>
> >    Related work on image pyramid and feature pyramid. In this paper, the data block preprocesses the images into multiple scales and the re-usable building block concatenates the features from multiple scales. A discussion/revisit on the literature about the image pyramid and feature pyramid is recommended here and it might give some inspirations to address the memory issues of training the model on Cityscapes.
>
> The memory issue is less related to the network architecture, but arises from the used deep learning framework. In TensorFlow, by default, operations like concatenation are not executed in-place, but introduce an additional node in the graph that causes an additional copy of the activations. The same problem also arises for other networks like, e.g., DenseNets. Fortunately, we could solve the memory issue and now present a full set of results for Cityscapes.

---

### Official Review · AnonReviewer3 · 2020-10-28
**Reivew**

**Rating:** 4
**Confidence:** 2

**Review:**

This paper proposes a homogeneous network structure, where a single building block is iteratively executed to refine the outputs, in order to achieve a good trade-off between latency and accuracy. Empirical studies are conducted to show the superiority of such homogeneous networks over ENet and CGNet on CamVid and Cityscapes datasets.

Concerns:

- The paper is not well organized which makes it hard to follow. The authors should revise the writing and clearly introduce the challenges, the contributions, the proposed architecture structure, the design insights, the experiment settings, and so on.

- The key concern about the paper is the lack of novelty and design insights. Why are homogeneous networks more beneficial for massively-parallel hardware systems? It would better to see the empirical evaluation and comparison against those non-homogeneous networks, such as multi-scale DenseNet, on a real hardware platform.

---

> ### Author Response · Authors · 2020-11-24
> **Addressing comments of reviewer 3**
>
> Thank you very much for these constructive, detailed and valuable comments that I will answer point by point below. For a summary of changes, see my comment above.
>
> >    The paper is not well organized which makes it hard to follow. The authors should revise the writing and clearly introduce the challenges, the contributions, the proposed architecture structure, the design insights, the experiment settings, and so on.
>
> We substantially revised the introduction and discussion to clarify the motivation, scope and conclusion. In addition, we improved the structure of our methods and experiment sections.
>
> >    The key concern about the paper is the lack of novelty and design insights. Why are homogeneous networks more beneficial for massively-parallel hardware systems? It would better to see the empirical evaluation and comparison against those non-homogeneous networks, such as multi-scale DenseNet, on a real hardware platform.
>
> We revised the introduction and discussion to better motivate the benefits of our approach. As also suggested by reviewer 1, we added an example to the discussion that showcases the benefit of our approach for current and future hardware systems. Since this work addresses hardware systems that are not easily accessible, yet, it is hard to show real-world numbers for the computational costs. A benchmark on CPUs, GPUs and TPUs could not show the benefit of the introduced network models, since their rather sequential processing mode is in contrast to the fully-parallel computation assumed by our model. We modified the discussion accordingly.

---

### Official Review · AnonReviewer2 · 2020-10-28
**Review opinion**

**Rating:** 3
**Confidence:** 3

**Review:**

This paper studies homogenious networks, which is defined by the paper as networks that reuse building blocks with shared or different weights multiple times during the inference of the network. During the inference, the network iteratively use the same set of blocks to process input feature maps with different resolutions, and each step, the output feature map can be used by the prediction head to generate the output. This paper studies the cost of the network, in terms of MACs, parameters, memory footprints, and the accuracy vs. the number of iterations. The author noted that for the studied network, they need to increase the MACs by 3x in order to match the performance of regular networks. Despite this, this kind of homogeneous networks can be useful for novel hardware architectures with limited memory bandwidth.

Strength of the paper: this paper presents an interesting study over a novel type of network and studied its accuracy-cost trade-off.

Weaknesses ofthe paper:
1/ The organization of the paper should be improved significantly. For example, the architecture of the homogeneous network is a central part of the paper, yet the introduction to it is only provided in two figures, and not in the method description.
2/ It is not clear what should be counted as the contribution of the paper. I am assuming a few possibilities: a) proposing the new homogeneous networks? However, there are already several previous papers mentioning this type of networks. [cite] b) Better performance achieved by using homogenous networks? However, the experiments of the paper show that the network under study is not as competitive as previous baselines cited in this paper. Also, though the authors suggest this type of network can be useful for future hardware processors, but without explicitly mentioning the processor, it is hard to justify this. c) The analysis of the homogeneous network and their accuracy/cost trade-off. This might be counted as a contribution, but it is not clear if this analysis can, say, help us improve the performance of homogeneous networks.

Overall, I think this paper is not complete and has not met the standard of acceptance.

---

> ### Author Response · Authors · 2020-11-20
> **References**
>
> Dear reviewer, regarding to your point 2a):
> Could you please point me to these previous papers that you mention.

---

> > ### Comment · AnonReviewer2 · 2020-11-23
> > **Relavent work**
> >
> > Please refer to the following papers that have adopted the homogenous networks (i.e., networks that repeatedly apply the same layer multiple times). This is an incomplete list, but it shows that the idea of homogenous network has been widely used prior to this work.
> >
> > [1] https://arxiv.org/abs/1909.01377
> > [2] https://arxiv.org/pdf/1801.10319.pdf
> > [3] https://arxiv.org/abs/1909.11942

---

> > > ### Author Response · Authors · 2020-11-24
> > > **Discussion of relevant work**
> > >
> > > Thank you very much for the quick reply and the interesting reads.
> > >
> > > * [1] Already in the initial version of this manuscript, we address the follow-up paper https://arxiv.org/abs/2006.08656 by the same authors that extends [1] to vision tasks. Although the network architecture looks quite similar, the motivation is different. [1] tries to match SOTA performance with their models. Our goal is to reduce the size of the computational graph as much as possible while matching the performance of efficient DNNs for semantic segmentation. Their smallest model has 7.8M parameters for Cityscapes while our model has 245k parameters. We revised the discussion to clarify this point. In addition, their method includes the raw image (8MB in float32 for Cityscapes) into the state of the iterative block, which prevents to shrink their networks to the regime of few billion MACs as shown by our method, for which the state has a size of only 4.8MB.
> > >
> > > * [2] In this work, a network building block is shared over different scales. This is similar to the work of Pinheiro & Collobert (ICML 2014) suggested by reviewer 4. The properties of their networks are very different from the scope of the study at hand. Their networks do not allow for anytime prediction and the computational graph is large due to the high resolution feature maps. We added the discussion of this reference to the manuscript.
> > >
> > > * [3] I consider this work to be clearly out of scope. First, this work addresses only NLP tasks, second the models are huge (18M to 233M parameters), and, third, transformer models require huge computational graphs.

---

> ### Author Response · Authors · 2020-11-24
> **Addressing comments of reviewer 2**
>
> Thank you very much for these constructive, detailed and valuable comments that I will answer point by point below. For a summary of changes, see my comment above.
>
> > 1/ The organization of the paper should be improved significantly. For example, the architecture of the homogeneous network is a central part of the paper, yet the introduction to it is only provided in two figures, and not in the method description.
>
> We intentionally placed the description of the model in the figure caption to provide the reader with nearby visual cues. However, we agree that the link to the main text is quite weak. We strengthened the link to the main text and introduced some redundancy to improve readability.
>
> > 2/ It is not clear what should be counted as the contribution of the paper. I am assuming a few possibilities:
> a) proposing the new homogeneous networks? However, there are already several previous papers mentioning this type of networks. [cite]
>
> Addressed in parallel thread.
>
> > b) Better performance achieved by using homogenous networks? However, the experiments of the paper show that the network under study is not as competitive as previous baselines cited in this paper. Also, though the authors suggest this type of network can be useful for future hardware processors, but without explicitly mentioning the processor, it is hard to justify this.
>
> We added an exemplary example to the discussion that should clarify the benefits of our introduced network model. See also our response to the comments of reviewer 1.
>
> > c) The analysis of the homogeneous network and their accuracy/cost trade-off. This might be counted as a contribution, but it is not clear if this analysis can, say, help us improve the performance of homogeneous networks.
>
> To our knowledge, this study is the first to introduce an iterative network with shared weights in the regime of few billion MACs for large-scale segmentation tasks. In addition, we set the state-of-the-art in terms of accuracy over the size of the computational graph. Only tiny-sized graphs fit onto massively-parallel hardware systems that allow for extremely high throughput. To emphasis these differences to existing work, we revised the manuscript and updated our list of contributions. In addition, we present ablation studies for the network training and the computational costs to obtain a deeper understanding where the benefits of this model come from and to sketch the implications for applications, respectively.

---

### Official Review · AnonReviewer4 · 2020-10-28
**Interesting idea; need more experimental justification**

**Rating:** 4
**Confidence:** 4

**Review:**

=========
Summary:

This paper proposes a homogeneous network structure for semantic segmentation, which optimizes for prediction accuracy, latency as well as memory footprint.  The paper studies anytime prediction setting and designs a re-usable single building block to reduce the memory footprint. Experimental results on CamVid data shows that it's possible to use a homogeneous network architecture to achieve competitive mIoU compared to previous work at the cost of increased MACs.  Experimental evaluation on larger datasets such as Cityscapes is prohibited due to memory constraints of the available GPUs.

=========
Pros:
*  The idea of using a homogeneous network is interesting and re-usable blocks are reasonable for the anytime prediction setting.
*  The paper is easy to follow.

=========
Concerns:

1.  Is the parameters of the re-usable building block updated iteratively during inference? The paper described two scenarios where the parameters of the re-usable building block can be updated or shared across multiple iterations.  My first question is,  is this for training time only or for both training and inference?

 a) At inference time if the parameter updates are allowed at each iteration, what is the mechanism to decide the new weights?  If an independent block was used and only the structure of the building block is reused, there would not be any benefit for latency or memory footprint.

 b) If the parameters of the building blocks are fixed during inference,  then it falls into the weight-sharing scenario where the model performance is much lower according to Figure 1 c).

2. Lack of anytime prediction experiments.  The paper claims it studies anytime prediction setting, however, I couldn't find explicit descriptions about it in the experiment section and descriptions about how the model choose the number of iterations during inference.

3. Missing Cityscapes experiments.  The paper claims in the abstract that the method was evaluated on cityscapes but in the discussion section, it says they couldn't provide cityscapes experiments due to memory issues.  The inconsistency in the paper should be fixed.

4. What is the exact benefit of the homogenous network?  The paper shows this structure requires at least 3 times larger MACs in order to achieve competitive performance. It also says this network has potential for novel, massively-parallel hardware accelerators.  This is a bit vague to me, could you provide a concrete example?

5. Related work on image pyramid and feature pyramid.  In this paper,  the data block preprocesses the images into multiple scales and the re-usable building block concatenates the features from multiple scales.  A discussion/revisit on the literature about the image pyramid and feature pyramid is recommended here and it might give some inspirations to address the memory issues of training the model on Cityscapes.

========

Overall, I think the idea of building a homogenous network is reasonable to me and may lead to a better trade-off between latency, accuracy, and memory. However, the current manuscript still needs much improvement.

---

> ### Author Response · Authors · 2020-11-24
> **Addressing comments of reviewer 4**
>
> Thank you very much for these constructive, detailed and valuable comments that I will answer point by point below. For a summary of changes, see my comment above.
>
> >    While I do believe that such a study could provide the community with a better understanding of architectures with re-used blocks, the scale of the study performed in this paper is too small to draw conclusions. The paper tackles a single task (semantic segmentation), and, more critically, evaluates a single architecture. There is therefore no evidence that the conclusions drawn here will generalize to other architectures/tasks, which significantly limits the potential impact of this paper on the community.
>
> We now provide a full set of results for the Cityscapes dataset. Additional tasks would certainly help to underline the benefits of re-using building blocks. However, we intentionally focus on semantic segmentation tasks, since these available scientific datasets have one of the largest spatial resolutions and, hence, are the most challenging use case. Different architectures are already covered to a certain extend by our grid search. What would you like to see beyond such a search? What additional insights do you expect?
>
> >    While I am not aware of similar studies, several important references tackling the task of recurrent semantic segmentation are missing, e.g., Pinheiro & Collobert, ICML 2014; Wang et al., ICCV 2019. These methods rely on different architectures, and studying the effect of parameter sharing in their frameworks would broaden the scope of this work.
>     Similarly, video segmentation has also been tackled with recurrent architectures, e.g., Ballas et al., ICLR 2016; Valipour et al., WACV 2017. Studying the use of parameter sharing in this context would thus also increase the potential impact of this work.
>
> Work in progress.
>
> >    While the results indeed evaluate the effect of parameter sharing on the chosen architecture, they seem to be somewhat disappointing in terms of absolute performance. In particular, on CityScapes (Fig. 1(b)), CNMM yields significantly higher mIoUs than the proposed method for the same number of MACs. Wouldn't it be possible to also extend this study to the CNMM architecture?
>
> We removed the comparison to CNMMs from the manuscript, since CNMMs do not allow for anytime prediction and now compare to the more competitive ESPNetv2. The size of the computational graph distinguishes our study from existing work. In terms of accuracy over the size of the computational graph, we set a new state-of-the-art.
>
> >    In the caption of Fig. 1, the authors mention that the parameters of the classification block are never shared. Does this mean that there is one classification block for each recurrent iteration?
>
> Yes. See also the tiny, but noticeable increase in parameters over iterations of ItNets in Fig. 5b. We revised the caption of Fig. 1.
>
> >    In Section 2.2, when explaining the normalization of the weight factors, I suggest using a different notation for the weights before and after normalization, e.g., w_n = w_n'/(\sum_i w_i').
>
> Done.
>
> >    The title of Section 3.1 (Neural Architecture Search) is a bit misleading, as the authors simply perform a grid search of a few hyper-parameters, not a proper search over a large space of different architectures as in the NAS sub-field.
>
> Good point, we renamed that section.
>
> >    The meaning of the different colors in Fig. 4 is not always explicitly defined.
>
> We revised the figure caption.

---

> > ### Author Response · Authors · 2020-11-24
> > **Discussion of proposed literature**
> >
> > > While I am not aware of similar studies, several important references tackling the task of recurrent semantic segmentation are missing, e.g., Pinheiro & Collobert, ICML 2014; Wang et al., ICCV 2019. These methods rely on different architectures, and studying the effect of parameter sharing in their frameworks would broaden the scope of this work.
> >
> > * In Pinheiro & Collobert (ICML 2014), a network building block is shared over different scales. This is similar to the work of  Cheng et al. suggested by reviewer 2. The properties of their networks are very different from the scope of the study at hand. Their networks do not allow for anytime prediction. We added the discussion of this reference to the manuscript.
> >
> > * Wang et al., ICCV 2019: Besides Bai et al, this work is closest to our study. So far, we only discuss it shortly, but will provide numerical comparisons and in-depth discussions, soon.
> >
> > > Similarly, video segmentation has also been tackled with recurrent architectures, e.g., Ballas et al., ICLR 2016; Valipour et al., WACV 2017. Studying the use of parameter sharing in this context would thus also increase the potential impact of this work.
> >
> > Both these works use recurrent units to connect intermediate feature maps of encoder and encoder-decoder networks over time, respectively. We discuss these works as relevant alternative techniques.

---

> > > ### Author Response · Authors · 2020-11-24
> > > **Recurrent U-Net**
> > >
> > > I added the reference numbers for recurrent U-Nets (Wang et al) that are significantly outperformed by ItNets.

---

### Author Response · Authors · 2020-11-24
**Revised version of the manuscript**

Thank you very much for these valuable and constructive comments. Since multiple reviewers asked for improvement in writing, a clearer motivation and more details about the impact on applications, we substantially revised the manuscript. Our motivation is now less hypothetical, but more driven by showcasing and exploiting the benefit of novel network models for execution on yet to be established fully-parallel hardware systems. I understand that actual results on such a hardware system would have been compelling, but this in my opinion a hen and egg problem and ideally both hardware and models are developed in parallel. We identified the size of the computational graph to be the enabler to fit a network model on these hardware systems that provide an extremely high throughput in this case. We adapted the title and abstract to reflect this revised motivation. We simplified the set of our experiments and completed the set of results for Cityscapes by adding data for iterative networks with weight sharing. We also added an exemplary application to the discussion and motivate why especially the task of semantic segmentation is suited as a benchmark. We do not compare to the CNMM anymore, since these models do not allow anytime prediction, and replaced them with the more competitive ESPNetv2 networks.

We adapted our manuscript to better match the common style of accepted publications in this conference by giving our network a catchy name, listing our contributions and shortening the figure captions.

Within the next days, we will address the references suggested by reviewer 2 and 4 and also compare our results to recurrent UNets. In addition, we will add the sizes of the graphs and latencies for the ENets and CGNets.

---

> ### Author Response · Authors · 2020-11-24
> **Added discussion of proposed literature**
>
> We added the discussion of all literature proposed by the reviewers.

---

> ### Author Response · Authors · 2020-11-25
> **Comparison to Recurrent U-Net and ENet**
>
> We now also compare our numbers to Recurrent U-Nets and ENets. In both cases, ItNets outperform these networks in mIoU over the size of the computational graph and in mIoU over MACs.

---

### Decision · Program_Chairs · 2021-01-07
**Final Decision**

**Decision:**

Reject

**Comment:**

All reviewers agree that the paper is well written and some of the experiments are interesting. However, the paper did not clearly highlight how this work fits in with prior research, neither did it show what the advantages of the presented homogeneous network are. The authors addressed some of these concerns in the rebuttal, but not enough to sway the reviewers. In the end all reviewers recommend rejection, and the AC sees no evidence to overturn this recommendation.